# LLM Query Scheduling with Prefix Reuse and Latency Constraints

**Gregory Dexter**
LinkedIn
gdexter@linkedin.com

**Shao Tang**
LinkedIn
shatang@linkedin.com

**Ata Fatahibaarzi**
LinkedIn
afatahibaarzi@linkedin.com

**Qingquan Song**
LinkedIn
ustcsqq@gmail.com

**Tejas Dharamsi**
LinkedIn
tdharamsi@linkedin.com

**Aman Gupta**
Nubank
aman.gupta@nubank.com.br

## Abstract

The efficient deployment of large language models (LLMs) in online settings requires optimizing inference performance under stringent latency constraints, particularly the time-to-first-token (TTFT) and time-per-output-token (TPOT). This paper focuses on the query scheduling problem for LLM inference with prefix reuse, a technique that leverages shared prefixes across queries to reduce computational overhead. Our work reveals previously unknown limitations of the existing first-come-first-serve (FCFS) and longest-prefix-match (LPM) scheduling strategies with respect to satisfying latency constraints. We present a formal theoretical framework for LLM query scheduling under RadixAttention, a prefix reuse mechanism that stores and reuses intermediate representations in a radix tree structure. Our analysis establishes the NP-hardness of the scheduling problem with prefix reuse under TTFT constraints and proposes a novel scheduling algorithm, $k$-LPM, which generalizes existing methods by balancing prefix reuse and fairness in query processing. Theoretical guarantees demonstrate that $k$-LPM achieves improved TTFT performance under realistic traffic patterns captured by a data generative model. Empirical evaluations in a realistic serving setting validates our findings, showing significant reductions in P99 TTFT compared to baseline methods.

## 1 Introduction

The rapid integration of large language models (LLMs) into online systems has spurred significant research aimed at improving their inference efficiency. Unlike traditional batch-processing environments, online usage demands a nuanced understanding of performance, prioritizing what is often termed "goodput" Zhong et al. [2024]—the maximum number of requests that can be served while meeting stringent constraints on time-to-first-token (TTFT) and time-per-output-token (TPOT). This shift in focus underscores the necessity of developing advanced inference and serving algorithms that optimize goodput, ensuring efficient and cost-effective deployment of LLMs in latency-sensitive applications.

39th Conference on Neural Information Processing Systems (NeurIPS 2025).

An important approach to improving LLM inference efficiency is prefix reuse. In autoregressive LLMs, prompts with shared prefixes can leverage the intermediate representations (i.e. key and value tensors, often stored in the key-value (KV) cache) of the common prefix, reducing both memory usage and computational overhead during inference. A notable implementation of this idea is RadixAttention, which stores the KV cache of processed queries and automatically reuses shared prefixes by maintaining a radix tree Zheng et al. [2024]. This tree tracks cached values and matches them with new incoming prompts, enabling efficient reuse without requiring manual intervention. To manage memory constraints imposed by the fixed size of the KV cache, the system employs a least recently used (LRU) eviction policy for cached values. This method is particularly impactful in scenarios where many prompts share prefixes, offering substantial savings in compute and memory. Moreover, its "out-of-the-box" functionality simplifies adoption by eliminating the need for users to manually analyze or encode prefix structures, making it a practical solution for real-world applications.

Automatic prefix reuse enabled by RadixAttention can substantially improve goodput in many realistic settings, particularly when there is large overlap among query prompts. However, under a stream of queries, the performance of RadixAttention becomes dependent on the order in which queries are processed (i.e., the "schedule"), and its implications for latency remain poorly understood.

The main scheduling algorithms considered in prior work are *First Come First Serve (FCFS)*, which processes queries in arrival order, and *Longest Prefix Match (LPM)*[1], which greedily maximizes KV cache reuse at each scheduling step. In the offline setting (where all queries are known in advance), Theorem 3.1 in Zheng et al. [2024] shows that LPM indeed maximizes cache reuse. However, the online setting, where queries arrive over time under tight time-to-first-token constraints, was not characterized.

In this work, we provide a theoretical exploration of RadixAttention scheduling in the online regime, focusing on non-preemptive scheduling. Our analysis and experiments show that existing scheduling approaches can lead to large TTFT spikes under heavy traffic, motivating a new scheduling algorithm that more robustly balances prefix reuse and waiting time. Specifically, we design a mechanism that exploits the benefits of LPM but mitigates its performance risks in high-load scenarios, resulting in superior performance for *long-prompt, short-output* (i.e., prefill-dominant) queries where Radix-Attention delivers significant efficiency gains. Examples of such applications include document summarization, coding assistants, and prompts with detailed instruction sets among others Anthropic [2024]. Recently, a well-studied generation use case that can benefit from faster prefill is the scaling of test-time compute (TTC) via *best-of-N* sampling, where $N$ outputs are generated in parallel and a verifier model is used to score them Snell et al. [2024], Cobbe et al. [2021]. TTC has enabled recent LLMs to dramatically improve performance on reasoning benchmarks for domains like coding and math Guo et al. [2025].

## 1.1 Contributions

• In Section 2, we introduce a formal model for analyzing the LLM query scheduling problem. Our model draws inspiration from prior work on the roofline model Imai et al. [2024] and incorporates experimental observations to ensure practical relevance. Despite its grounding in real-world considerations, the model remains sufficiently simple to facilitate analytical insights, including a formal specification of the "query stream" (Definition 1) and a computational model for LLMs (Definition 2), enabling further exploration of the problem.

• We show that the decision problem of determining whether a TTFT constraint can be satisfied for a given query stream is NP-Hard when using Radix Attention (Theorem 1). This is in contrast to the case without Radix Attention, where latency is trivially minimized by FCFS, or the case with Radix Attention and uniform arrival times, where latency is minimized by LPM Zheng et al. [2024].

• Although the decision problem is NP-hard, we introduce a data generative model (see Definition 4) that effectively captures the behavior of realistic query streams in key applications. Additionally, we present a generalized algorithm, $k$-LPM (Algorithm 1), which outperforms both FCFS and LPM under this data generative model (see Theorem 2 and Corollary 3).

---

[1]LPM is the default query scheduling algorithm in SGLang v0.4.1. Meanwhile, FCFS is an option in SGLang and the default scheduling algorithm in vLLM v0.6.6.

- We validate our theoretical results with experiments demonstrating that our predictions hold in practice by running a Llama-3.1-8B-Instruct model on the SGLang serving framework using real prompts. In particular, the $k$-LPM algorithm is able to attain reduced P99 TTFT across a range of request rates and settings of the hyperparameter $k$.

- Finally, in Appendix A we prove that an approximation algorithm exists that, for a length $n$ query stream $\mathcal{Q}$ and TTFT constraint $T$, either 1) certifies no schedule exists satisfying the TTFT constraint, or, 2) returns a schedule such that the $(1 - p)$-th percentile TTFT is at most $T$ for $p \in (0, 1)$. Furthermore, this algorithm runs in $\mathcal{O}(n \cdot \exp(1/p \log 1/p))$ time.

## 1.2 Related work

There has been significant interest in prior work on scheduling LLM queries in order to maximize throughput while satisfying TTFT constraints. Some examples are FastSwitch, which employs a priority-based scheduler with preemption to dynamically allocate resources to effectively reduce TTFT and GPU idleness Shen et al. [2024], and Orca, which employs iteration-level scheduling, processing each model iteration separately allowing for flexible batching and immediate response to newly arrived requests Yu et al. [2022]. Many other papers increase throughput subject to latency constraints by innovations to the processing schedule of incoming prompts Zhong et al. [2024], Patel et al. [2024], Jain et al. [2024], Agrawal et al. [2024]. Collectively, this prior work underscores the importance of dynamic scheduling in achieving high throughput without compromising latency guarantees.

The most relevant work to ours addresses scheduling of LLM queries with consideration of prefix reuse. Zheng et al. [2024] introduced RadixAttention and the LPM scheduler, which we build upon, but their work provides only limited exploration of scheduling strategies. Srivatsa et al. [2024] proposes a priority-based local scheduler in Section 3.3 aimed at balancing prefix reuse and waiting times. The scheduler works by assigning requests to priority groups based on prefix cache hit rate and then selects a number of prompts in each group proportionally to its priority, but no accompanying analysis is provided. Qin et al. [2024] integrates RadixAttention in local instances but primarily emphasizes maintaining a disaggregated KV cache to balance the decode and prefill phases, enhancing throughput while adhering to latency constraints in highly overloaded scenarios.

Analytic exploration of LLM inference efficiency is relatively limited, with some notable examples. Kim et al. [2024] propose INFERMAX, an analytical framework for evaluating schedulers and deriving theoretical performance bounds, while identifying prefix-sharing awareness as a future direction. Yang et al. [2024] analyze the behavior of LLM queries using an M/G/1 queue while accounting for unknown decoding length, and Guldogan et al. [2024] examine multi-bin batching to boost throughput. These studies underscore the value of theoretical approaches, which our work advances through a focus on prefix-aware scheduling.

## 1.3 Notation

Let bold lower case letters denote strings, e.g., $\mathbf{x}$ or $\mathbf{y}$, over some fixed alphabet. Let $|\mathbf{x}|$ denote the length of string $\mathbf{x}$. Let $\text{Overlap}(\mathbf{x}, \mathbf{y})$ denote the length of the maximal prefix overlap between $\mathbf{x}$ and $\mathbf{y}$.

## 2 Computational model of RadixAttention

In this section, we provide a model of LLM computation that allows for theoretical study of RadixAttention under different scheduling algorithms. Consider a single LLM instance (that may be on a single GPU or parallelized across multiple GPUs in various ways). This LLM instance can process queries using continuous batching. The time for a single pass can be understood from the "roofline model" (see Appendix A.2 of Imai et al. [2024] for details on applying the roofline model to LLM inference).

First, we define the query stream as a collection of prompts with associated arrival times.

**Definition 1.** *(Query stream) Let a query stream of length $n$ be denoted by $\mathcal{Q} = (\mathbf{x}_i, t_i)_{i \in [n]}$, where $\mathbf{x}_i$ is an arbitrary length string in some fixed alphabet[2] and $t_i \geq 0$ is the arrival time of the $i$-th query.*

---

[2]This alphabet will be the set of tokens in practice.

Note that this definition specifies a fixed finite collection of queries rather than a distribution as is common in queuing theory. We make this choice to simplify the analysis under the added complexity of prefix reuse, and to enable focus on the "burst traffic" behavior of TTFT rather than stable state behavior of the queue. That is, we are most interested in the behavior of scheduling algorithms in the periods of time where queries arrive faster than they can be processed and so the queue becomes temporarily long.

We present a formal model for the computation time of an LLM operating with a batch size of one under a specified queue ordering. This model captures both the time to process each prompt—accounting for prefix reuse from the preceding query—and the constraint that processing cannot start before a prompt's arrival time. We focus on scenarios where the prefill stage constitutes the primary contributor to total inference time, and where RadixAttention provides the largest improvement within that stage.

**Definition 2** (LLM Instance Computation). *Consider a stream of queries $\mathcal{Q} = \{(\mathbf{x}_i, t_i)\}_{i \in [n]}$. Suppose these queries are processed in the order $\mathbf{x}_{j_1}, \mathbf{x}_{j_2}, \ldots, \mathbf{x}_{j_n}$. Let $R(j_k)$ denote the completion time of the $j_k$-th query. Then $R(j_1) = |\mathbf{x}_{j_1}| + c_{\mathrm{attn}}|\mathbf{x}_{j_1}|^2$, and,*

$$R(j_k) = \max\{R(j_{k-1}), t_{j_k}\} + (1 + c_{\mathrm{attn}} \cdot |\mathbf{x}_{j_k}|)\big(|\mathbf{x}_{j_k}| - \mathrm{Overlap}(\mathbf{x}_{j_k}, \mathbf{x}_{j_{k-1}})\big).$$

*In other words, the $j_k$-th query cannot start processing until its arrival time $t_{j_k}$. Its processing cost is proportional to the prompt length minus the prefix it shares with the previous query. We assume the cache is empty initially, so the first query costs $|\mathbf{x}_{j_1}|$.*

During inference, the prefill stage involves a forward pass over all input tokens. The above formula for $\mathcal{R}(j_k)$ captures the dominant computational cost of a forward pass through an autoregressive transformer architecture. Layers such as the query, key, and value projections or MLP modules scale linearly with the number of tokens that must be processed in a query. Meanwhile in the attention mechanism, each token must attend to all previous tokens. Taking into account the cached prefix, this step scales with $|\mathbf{x}_{j_k}|(|\mathbf{x}_{j_k}| - \mathrm{Overlap}(\mathbf{x}_{j_k}, \mathbf{x}_{j_{k-1}}))$. For short to medium sequence lengths, the point-wise feed-forward network (FFN) typically dominates computation time. For long sequence lengths, self-attention dominates, as it scales quadratically. We capture the relative cost of the MLP versus self-attention operations for a fixed architecture by the $c_{\mathrm{attn}}$ constant. By focusing on a prefill-dominated regime, we simplify the analysis while retaining real-world relevance.

## 2.1 Batch size and prefix cache

The original specification of RadixAttention does not leverage reuse of prefixes of prompts in a single batch (see Algorithm 1 in Zheng et al. [2024]). However, more recent development in SGLang has enabled within batch prefix sharing. In this case, behavior of scheduling algorithms in the batched setting can be closely approximated by the batch size one setting in Definition 2. Computing a query processing order and then dividing the queries into $B$ sized bins sequentially provides a schedule for the $B$ batch size setting that attains the same amount of prefix reuse. The only difference is that the time that the $i$-th query is finished being processed will be the same as when the $\lceil i/B \rceil$-th query is finished being processed. This difference will be negligible when the query arrival rate is high relative to the batch size.

A discrepancy between Definition 2 and the behavior of RadixAttention is that our model assumes only the last processed prompt is cached, whereas RadixAttention allocates a fixed amount of memory for caching. We choose to make this simplification to avoid introducing an additional hyperparameter and because the behavior remains similar under standard settings. Specifically, we expect the behavior to be similar when prompt length is fairly uniform and the query arrival rate is high relative to the TTFT constraint.

In our experiments (Section 5), we use dynamic batch size and a fixed memory pool for the prefix cache in the SGLang framework and observe that our theoretical predictions based on Definition 2 are still accurate.

## 3 Complexity of scheduling under TTFT constraints

We first explore the complexity of the decision problem for determining whether there exists a schedule for a fixed queue that satisfies a given TTFT constraint $T$ for each query. By reduction to

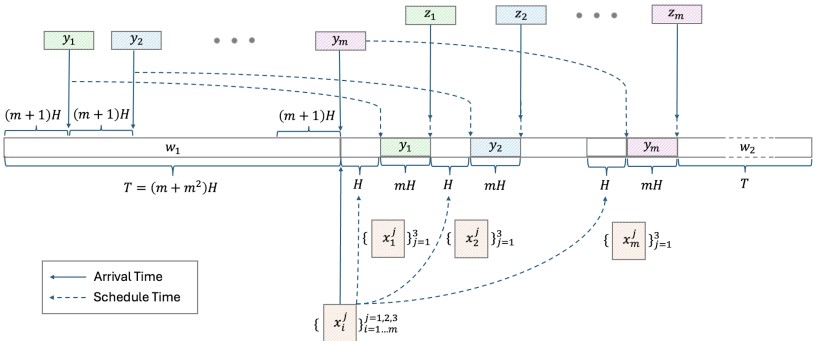

Figure 1: This figure graphically represents the imposed structure for any feasible schedule in the query stream construction of Theorem 1. Note that the only flexibility in the schedule is how the set of strings $\{\mathbf{x}_i\}_{i \in [3m]}$ fits into the $m$ time windows of size $H$. The solid lines represent arrival times and the dashed lines represent processing start times.

the `3-PARTITION` problem, we show that it is NP-Hard to decide feasibility of a queue under a TTFT constraint, despite this problem being trivial in the case where there is zero prefix reuse (where FCFS is optimal) or in the offline setting where all arrival times are uniform (where LPM is optimal).

## 3.1   NP-hardness of feasibility determination

We provide the definition of the `3-PARTITION` problem, which is NP-Hard in the strong sense. See SP15 in the appendix of Garey and Johnson [1979] for reference.

**Definition 3.** *(3-PARTITION) Let $m \in \mathbb{N}$ and $H > 0$. Given a set $\mathcal{A}$ of $3m$ integers such that $H/4 < a < H/2$ for all $a \in \mathcal{A}$ and $\sum_{a \in \mathcal{A}} a = mH$, decide if $\mathcal{A}$ can be partitioned into $m$ disjoint sets $\mathcal{A}_1, ..., \mathcal{A}_m$ such that $\sum_{a \in \mathcal{A}_i} a = H$ for all $i \in [m]$.*

The intuition behind the proof of Theorem 1 is as follows. The introduction of prefix reuse along with non-uniform arrival times allows us to construct query pairs that must be processed at a particular time. As an example, let query $\mathbf{x}$ arrive at time $H$ and query $\mathbf{y}$ arrive at $T + H$ where $\mathbf{x}$ and $\mathbf{y}$ have the same prompt. In order to achieve prefix reuse between these two prompts without any idle time while satisfying the TTFT constraint $T$, $\mathbf{x}$ must finish being processed at $T + H$ and $\mathbf{y}$ must start being processed at this time. By using this idea to introduce constraints on processing times, we may construct $m$ "windows" of size $H$ that a set of prompts may be feasibly scheduled within. By then constructing a set of queries with processing time equal to the integers in Definition 3, deciding the existence of a feasible schedule solves the `3-PARTITION` problem.

**Theorem 1.** *Deciding if there is a processing order in query stream $\mathcal{Q}$ (Definition 1) such that a TTFT constraint $T$ is satisfied under the computational model of Definition 2 is an NP-Hard problem.*

*Proof.* Let $\mathcal{A}$ be an instance of the `3-PARTITION` problem defined in Definition 3. We construct a query stream $\mathcal{Q}$ such that determining whether the queries can be scheduled to meet the TTFT constraint $T = (m + m^2)H$ under the computation model of Definition 2 is equivalent to deciding the `3-PARTITION` instance.

*Query construction.*

Define:

- $\mathcal{X}$: A set of $3m$ queries $\{\mathbf{x}_i\}$, where each prompt consists of a unique character, leading to zero prefix overlap between queries. Furthermore, $|\mathbf{x}_i| = a_i$ for some indexing that matches $\mathcal{A}$. Assign all $\mathbf{x}_i$ the same arrival time $t = (m + m^2)H$.
- $\mathcal{Y}$: An ordered set of $m$ queries $\{\mathbf{y}_i\}$, each of length $mH$, each composed of a unique character not appearing in $\mathcal{X}$. Assign arrival time $t_i = i(H + mH)$ to $\mathbf{y}_i$.
- $\mathcal{Z}$: A set of $m$ queries $\{\mathbf{z}_i\}$ where each $\mathbf{z}_i$ is identical to $\mathbf{y}_i$ in content but arrives at time $t_i = T + i(H + mH)$.
- $\mathbf{w}_1$ and $\mathbf{w}_2$: Two additional queries, each of length $T$, composed of characters distinct from those in $\mathcal{X} \cup \mathcal{Y}$. Let $\mathbf{w}_1$ arrive at $t = 0$ and $\mathbf{w}_2$ arrive at $t = 2T$.

Set the overall query stream to be $\mathcal{Q} = \mathcal{X} \cup \mathcal{Y} \cup \mathcal{Z} \cup \{\mathbf{w}_1, \mathbf{w}_2\}$. Observe:

1. $\mathbf{w}_1$ must begin processing at time $t = 0$. Since $|\mathbf{w}_1| = T$ and there is no prior cache, it finishes exactly at $t = T$.
2. Since $\mathbf{w}_2$ composed of characters distinct from those in $\mathcal{X} \cup \mathcal{Y}$, $\mathbf{w}_2$ has zero cache overlap when it arrives at $t = 2T$ [3]. Then, $\mathbf{w}_2$ must begin processing exactly at $t = 2T$ to finish by $3T$, thereby forcing *all* queries in $\mathcal{X} \cup \mathcal{Y} \cup \mathcal{Z}$ to be completed within the time window $[T, 2T)$.

Within $[T, 2T)$, the only possible nontrivial cache overlaps come from pairs and $(\mathbf{y}_i, \mathbf{z}_i)$ because all other queries have prompts with distinct characters. For each $i \in [m]$, the maximal prefix overlap between $\mathbf{y}_i$ and $\mathbf{z}_i$ reduces the processing time by $|\mathbf{y}_i|$. However, to fit all queries from $\mathcal{X} \cup \mathcal{Y} \cup \mathcal{Z}$ into $[T, 2T)$, *every* possible overlap must be fully utilized. This rigid constraint implies:

- The total time to process $\mathcal{Y}$ and $\mathcal{Z}$ (accounting for maximum $\mathbf{y}_i$–$\mathbf{z}_i$ overlap) is: $\sum_{\mathbf{y}_i \in \mathcal{Y}} |\mathbf{y}_i| = m^2 H = T - mH$.
- Processing must then be continuous (no idle time) in $[T, 2T)$, and must attain maximum overlap between every $\mathbf{y}_i$ and $\mathbf{z}_i$. In turn, this implies that $\mathbf{y}_i$ must finish being processed at exactly $T + i(H + mH)$ so that $\mathbf{z}_i$ can be processed immediately as it arrives without violating the TTFT constraint on $\mathbf{y}_i$.

*Partition of $\mathcal{X}$ and relation to `3-PARTITION`.*

We next show that $\mathcal{X}$ must be partitioned into $m$ disjoint batches, each of total length $H$. Observe that for each $i \in [m]$, since $\mathbf{z}_i$ must start right when $\mathbf{y}_i$ finishes, we have

$$\mathbf{y}_i \text{ end (and thus) } \mathbf{z}_i \text{ begin at } T + i(H + mH).$$

Then $\mathbf{y}_{i+1}$ must be processed starting from $T + i(H + mH) + H$ in order to finish by $T + (i + 1)(H + mH)$ and maintain the "no idle time" schedule. Since $\mathbf{z}_i$ is identical to $\mathbf{y}_i$, its processing time is zero in this case. Therefore, between $\mathbf{z}_i$ finishing at $T + i(H + mH)$ and $\mathbf{y}_{i+1}$ starting at $T + i(H + mH) + H$, there is exactly a length-$H$ sub-interval available. Since $\sum_{\mathbf{x}_i \in \mathcal{X}} |\mathbf{x}_i| = mH$, the only way to fill these $m$ sub-intervals continuously is to divide $\{\mathbf{x}_i\}$ into $m$ disjoint groups, each summing to exactly $H$. But deciding such a partition is precisely the `3-PARTITION` problem. Consequently, scheduling $\mathcal{Q}$ to meet the TTFT constraint is possible if and only if the instance $\mathcal{A}$ of `3-PARTITION` admits a feasible partition. □

In Appendix A, we further explore the general problem of scheduling queries to satisfy a TTFT constraint. In Theorem 4, we prove that there exists an algorithm that accepts a query stream and TTFT constraint $T$ and either certifies that there is no schedule satisfying constraint $T$ or returns a schedule such that the $(1 - p)$-th percentile TTFT is at most $T$ in $\mathcal{O}(n \cdot \exp(1/p \log 1/p))$ time.

## 4  $k$-LPM scheduling algorithm

In this section, we introduce our proposed scheduling algorithm, $k$-LPM, that generalizes the FCFS and LPM scheduling algorithms. Note that it reduces to LPM when $k = \infty$ and reduces to FCFS when $k = 1$.

Below, we show that $k$-LPM can achieve superior performance in terms of TTFT on random queues under a data generative model that captures the relevant properties of realistic use cases for RadixAttention (Theorem 2). This shows that, despite the intractability of the general scheduling problem, it is still possible to obtain theoretically grounded improvement over existing methods in realistic settings. In Section 5, we further support this improvement with experiments showing that the $k$-LPM algorithm achieves better TTFT performance than FCFS or LPM on queues constructed using real prompt distributions.

The intuition behind Algorithm 1 is that it first performs $k$ greedy prefix-match steps in the spirit of LPM to maximize prefix reuse. After these $k$ steps, it processes the oldest query in the queue, mirroring FCFS. This strategy circumvents the LPM failure case, where a query could be unprocessed if its prompt never have sufficiently high prefix overlap. At the same time, it retains the significant prefix-reuse advantage that LPM provides.

---

[3]It is trivial to show that some queries must be processed between $\mathbf{w}_1$ and $\mathbf{w}_2$ in order to meet the TTFT constraint.

**Algorithm 1** $k$-LPM

---

1: **Input:** Input queue of prompts and arrival times $\mathcal{Q} = (\mathbf{x}_i, t_i)$
2: **while** true **do**
3:    Process the oldest query, i.e., $\mathbf{x}_i$ such that $i = \operatorname{argmin}_j t_j$
4:    **for** $i = 1, \ldots, k - 1$ **do**
5:       Process $\mathbf{x}$ that maximizes the prefix cache hit rate
6:    **end for**
7: **end while**

---

## 4.1 Data generative model

Although deciding whether a TTFT constraint can be satisfied for a given query stream is NP-Hard (see Theorem 1), we are able to show that under a data generative model capturing properties of practical use cases, $k$-LPM achieves an improvement on the maximum TTFT. Our data generative model has the following additional structure.

**Tree structured queries:** Recall that the maximum prefix reuse is attained for a fixed set of prompts by DFS traversal of the radix tree constructed from all prompts. Hence, RadixAttention can only provide significant efficiency gains if the sum of edges of the radix tree constructed from prompts in a query steam is significantly less than the sum of prompt lengths. Fortunately, many applications of LLMs fulfill this assumption.

In this section, we restrict our attention to instances where queries in a queue approximately follow a tree structure of low height. An example of such prompt structure used in Team [2025] is $\mathbf{x} =$ (base_prompt)(user_context)(doc). Here, all queries share the same (base_prompt), and multiple queries may share the same (user_context).

Examples of such structures include generative usecases: personalized content generation Zhang et al. [2024], conversational context-aware question answering Zaib et al. [2022], and the predictive usecase with LLMs as engagement predictors in recommendation systems Wu et al. [2024]. These scenarios exemplify applications where the prompt structure remains consistent while the user context varies, allowing for efficient processing and relevant responses. By focusing on such structured instances, we can better analyze and optimize the computational models for LLMs under constrained scheduling conditions.

Our data generative model considers the simplest case of prompts constructed from a height two prefix tree, where the edges at each depth are constant. The arrivals of the queries are regular, but the order of the arrivals is sampled uniformly from all permutations of the queue. This is the simplest model that captures the interplay between the tree structure of the prompt prefixes and the arrival rate of the queries under randomness in the queue arrival order. For ease of exposition, we keep with the (user) and (doc) terminology of the previous example use case. However, these ideas apply generally to query streams with approximately tree-structured prompts.

**Definition 4** (Regular Arrival Shuffled Queue). *Let $n, u, k, d, s \in \mathbb{N}$ be parameters such that $k$ divides $n$. We form a collection of $n$ queries, each denoted by (user)(doc), where:*

- *(user) is a substring of length $u$, repeated in exactly $k$ distinct queries.*
- *(doc) is a substring of length $d$, unique to each query.*
- *Each (user) and (doc) starts with a distinct character from a large enough alphabet, ensuring zero overlap among different (user) or (doc) substrings.*

*Thus, there are $n/k$ distinct user substrings, each used $k$ times, and $n$ distinct doc substrings, one per query, yielding $n$ total prompts. We construct the regular arrival shuffled queue $\mathcal{Q}_n = \{(\mathbf{x}_{\sigma(i)}, s \cdot \sigma(i)) \mid i \in [n]\}$ by sampling a permutation $\sigma$ uniformly randomly from the symmetric group $S_n$, and assigning arrival time $s \cdot \sigma(i)$ to the $i$-th prompt.*

The structural assumptions in Definition 4 can certainly be relaxed. In real settings, there would likely be negligible but non-zero overlap between unique user and documents, and the repitions of each user may not be uniform. Here, we avoid these details to focus on clarity regarding the most pertinent structure.

## 4.2 TTFT improvement from $k$-LPM

In scenarios where each query can be processed swiftly—specifically, when the processing time of a given query is less than the inter-arrival interval $s$—the FCFS scheduling algorithm is optimal. However, in more practically relevant burst-traffic regimes where queries arrive in rapid succession and cannot be processed fast enough, a backlog of unprocessed queries inevitably forms. To illustrate this, consider a toy example with $c_{\text{attn}} = 0$, $n = 4$ queries, a replication factor $k = 2$, and parameters $u = 5$ and $d = 5$. The queries are denoted as $\mathbf{x}_1 = (\texttt{user})_1(\texttt{doc})_1$, $\mathbf{x}_2 = (\texttt{user})_2(\texttt{doc})_2$, $\mathbf{x}_3 = (\texttt{user})_1(\texttt{doc})_3$, and $\mathbf{x}_4 = (\texttt{user})_2(\texttt{doc})_4$. In the case where the inter-arrival time $s = 10$, FCFS scheduling is clearly optimal, resulting in a uniform TTFT of 10 units for each query. Conversely, in the case with $s = 0$, representing a burst-traffic scenario, the LPM scheduling strategy becomes optimal. Under FCFS, the processing order is $\mathbf{x}_1$, $\mathbf{x}_2$, $\mathbf{x}_3$, $\mathbf{x}_4$ and $\texttt{TTFT}_i = 10 \cdot i$ with $\max(\texttt{TTFT}_i) = \texttt{TTFT}_4 = 40$. Under LPM, the processing order is rearranged to $\mathbf{x}_1$, $\mathbf{x}_3$, $\mathbf{x}_2$, $\mathbf{x}_4$ with TTFTs of 10, 15, 25, and 30, and $\max(\texttt{TTFT}_i) = 30$. This example underscores the improvement of TTFT from cache reuse in practical scenarios.

In the following theorem, we show that $k$-LPM has a lower maximum TTFT than FCFS or LPM[4] with high probability on instances of the regular arrival shuffled queue (Definition 4). We set the hyperparameter $k$ in $k$-LPM to match the number of user repetitions defined in Definition 4 for simplicity. In practice, the hyperparameter $k$ in $k$-LPM can be determined through back-testing or by employing an adaptive multi-armed bandit approach to achieve better performance than the provable improvement observed in the simple setting. In our experiments (Section 5), we empirically measure the performance for varying values of the hyperparameter $k$.

Intuitively, the theorem shows that when $s$ is small and $u$ is large, LPM is much better than FCFS, and $k$-LPM retains this advantage. On the other hand, when $s$ is relatively large, FCFS is better and $k$-LPM retains a $\frac{1}{k}$ factor of the $sn$ reduction in TTFT. For intermediate values of $s$ and $u$, $k$-LPM is better than both algorithms as we prove in Corollary 3. Proofs for these results are provided in Appendix B.

**Theorem 2** (LPM/FCFS vs. $k$-LPM). *Let $\mathcal{Q}_n$ be a regular arrival shuffled queue (Definition 4) of length $n$ with $k$ repetitions of each user prefix, user history length $u$, document length $d$, and inter-arrival gap $s$. Suppose the queue starts being processed at time $T \geq s\,n$, and let $\texttt{TTFT}_i$ denote the time-to-first-token of the $i$-th query under a specified scheduling algorithm. Then, under the computational model of Definition 2 with $c_{\text{attn}} = 0$:*

- *(LPM) For every $\epsilon > 0$ and $\delta \in (0,1)$, there is an $n_0$ such that for all $n \geq n_0$, with probability at least $1 - \delta$ (with respect to random shuffle and randomness in LPM):*
  $$\max_{i \in [n]} \texttt{TTFT}_i \;\geq\; T \;+\; (1-\epsilon)\, n \left( \frac{u}{k} + d \right).$$

- *(FCFS) For every $\epsilon > 0$ and $\delta \in (0,1)$, there is an $n_0$ such that for all $n \geq n_0$, with probability at least $1 - \delta$ (over the random shuffle):*
  $$\max_{i \in [n]} \texttt{TTFT}_i \;\geq\; T \;+\; (1-\epsilon)\, n \left( u + d - s \right).$$

- *($k$-LPM) Deterministically (i.e. for any arrival order), Algorithm 1 satisfies:*
  $$\max_{i \in [n]} \texttt{TTFT}_i \;\leq\; T \;+\; n \left( \frac{u}{k} + d - \frac{s}{k} \right).$$

**Corollary 3.** *For any values of $s$, $k$, $u$, and $d$ such that $0 < s < u$ and $k > 1$, and for any $\delta \in (0,1)$, there exists $n_0 \in \mathbb{N}$ such that $k$-LPM achieves a lower maximum TTFT that LPM and FCFS simultaneously on the regular arrival shuffled queue (Definition 4) for any value of $n \geq n_0$ with probability at least $1 - \delta$. This result holds for any value of $c_{\text{attn}} \geq 0$ in the computational model of Definition 2.*

## 5 Experiments

In this section, we measure the performance of $k$-LPM versus FCFS and LPM in a realistic setting. Our results validate the predictive power of the computational model from Section 2 and the data generative model from Definition 4 in real-world serving scenarios.

---

[4]We assume that ties in prefix overlap are broken by uniform sampling in the LPM algorithm.

In our experiments, we use the Llama-3.1-8B-Instruct model Dubey et al. [2024] with tensor paralelism across eight A100 GPUs. We run the experiment using the SGLang v0.4.1 serving framework. In particular, we evaluate the timing metrics using SGLang's serving benchmark utility SGLang [2024] and only modify the benchmarking dataset. We implement the $k$-LPM algorithm as an extension to the current LPM implementation in SGLang. Finally, we construct the dataset used for benchmarking by sampling four prompts with shared user history from the 8k context length prompts described in Team [2025] for 2100 prompts in total. We follow the textual interface of Team [2025]—(Instruction, Member Profile, Past Interactions, Question)—where the first three constitute the shared prefix and only the Question varies; see 360Brew, Table 1 for an example of this exact layout. We then randomly shuffle the ordering of these prompts and use 100 for warm up of the benchmarking server.

## 5.1 Performance of $k$-LPM

We measure P99 TTFT vs. request rate for varying values of $k$ in the $k$-LPM algorithm. Our key observation (Figure 2) is that setting the hyperparameter to $k = 2$ achieves reduced P99 TTFT across a wide range of request rates compared to FCFS and LPM. Note that SGLang's benchmarking utility uses a Poisson arrival process, so "request rate" refers to the average number of requests per second. Additional experiment results are provided in Appendix C showing other TTFT percentile metrics and performance when varying the amount of prefix reuse.

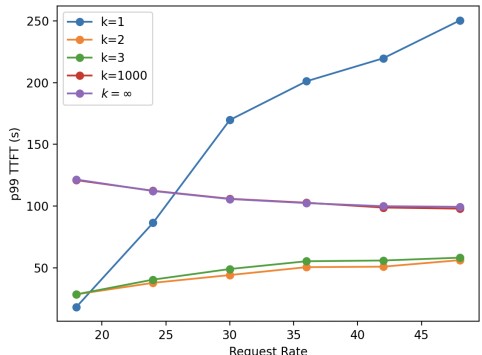

Figure 2: We measure P99 TTFT versus request rate for five values of the hyperparameter $k$ on 2000 randomly shuffled prompts from the usecase described in Team [2025]. Note that $k = 1$ corresponds to FCFS and $k = \infty$ corresponds to LPM.

The experimental results not only highlight the benefits of the $k$-LPM scheduling algorithm but also demonstrate that our theoretical framework—as encapsulated by Theorem 2—accurately predicts scheduling behavior in real-world settings. This holds true even though the experiments do not strictly adhere to all of the assumptions required by the theorem. In particular, we make the following observations:

- **$k$-LPM consistently outperforms both FCFS and LPM.** Over a wide range of request rates considered, $k$-LPM achieves a lower TTFT. This underscores its robust advantage, especially under moderate to heavy loads. Additionally, we note that large values of $k$ (e.g., $k = 1000$) behave like $k = \infty$.

- **FCFS shows better performance at low request rates, while LPM is advantageous at higher rates.** This matches our theoretical insight in Theorem 2, where a larger inter-arrival time $s$ favors FCFS, but as $s$ decreases (i.e., the request rate grows), LPM becomes more efficient than FCFS.

- **$k$-LPM performance remains robust under realistic models and prompt data, even when theoretical assumptions are relaxed.** Despite using a real LLM, an off-the-shelf serving framework, and an industrial prompt dataset, the observed scheduling behavior closely matches our theory. Here, prefix overlap structure is only approximately tree-like and random tie-breaking holds in practice; streaming processing (starting on first arrival) shows the "$T \geq sn$" assumption is inconsequential; and even when $k$ differs from the true replica count (we used four replicas), choices like $k = 2$ or $3$ still yield substantial P99 TTFT improvements.

## 6 Future work

The primary objectives of this work were to formalize the LLM query scheduling problem in the context of RadixAttention and to develop a practical scheduling algorithm for latency-sensitive applications. This theoretical framework not only informs the design of new methods, but is particularly

valuable given the current lack of empirical benchmarks for comparing scheduling strategies. However, further advancements in this area will necessitate empirical evaluation using real-world arrival patterns. In particular, the described behavior depends on the complex interaction between constant factors dictating the arrival rate, query processing rate, and average prefix reuse, along with batch size, radix tree memory limit, and other factors in real serving frameworks. Greater understanding of LLM query scheduling in real systems necessitates a standardized, measurement-driven evaluation suite to characterize the practically relevant regime.

The proposed problem formulation and theoretical results leave many interesting extensions open. One open question is whether there is an algorithm which returns a schedule satisfying a constraint on the $(1-p)$-th percentile TTFT in $\mathrm{poly}(1/p)$ time, i.e., a polynomial time approximation scheme. Additional directions of interest would be extending the computational model of Section 2 to handle distributional query streams or non-constant decoding length. Finally, generalizing the data generative model of Definition 4 may be interesting to capture other properties of real data.

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

# A  Percentile TTFT constraint approximation

The hardness result of Theorem 1 motivates us to consider approximation guarantees for computing a schedule satisfying constraints on the TTFT. One important relaxation of the problem is to instead consider scheduling algorithms that satisfy a constraint on a fixed percentile of the per-query TTFTs. This problem is of interest since latency constrained applications typically seek to bound the P95 or P99 latency of a response in practice. In this section, we show that such a relaxation is tractable when the interaction distance of queries in a queue (controlled by the latency constraint $T$ and maximum prompt length $m$) is bounded.

Theorem 4 proves there is an algorithm that runs in polynomial time with respect to the query stream length that either returns a schedule satisfying constraint $T$ on the $(1-p)$-th percentile TTFT or certifies that no feasible schedules exists such that all queries satisfy the TTFT constraint $T$. Note that there is a necessary gap where the the positive case pertain the the percentile TTFT problem and the negative case certifies no schedule satisfies a constraint on the maximum TTFT. This gap is fundamental, since otherwise we could solve the problem for the maximum TTFT in by adding enough unsatisfiable queries and solving the percentile TTFT decision problem.

**Theorem 4.** *There exists an algorithm that takes a length $n$ query stream $\mathcal{Q}$ (see Definition 1) and a TTFT constraint $T > 0$ as input and either:*

1. *Returns a certificate that no schedule exists for $\mathcal{Q}$ that satisfies TTFT constraint $T$, or*

2. *Returns a schedule for $\mathcal{Q}$ such that the $(1-p)$-th percentile TTFT is at most $T$,*

*under the computational model of Definition 2 with $c_{\mathrm{attn}} = 0$. Furthermore, if every query is at most length $m$ and no query is an exact prefix of another, then this algorithm runs in $\mathcal{O}(n \cdot \exp(1/p \log 1/p))$ time when $m$ and $T$ are bounded by a constant.*

*Proof.* At a high level we will prove that, if there exists a schedule for $\mathcal{Q}$ satisfying the TTFT constraint $T$ for all queries, then there exists a subset $\mathcal{Q}' \subset \mathcal{Q}$ of size $(1-p)n$ that also satisfies the constraint $T$ for all queries. By the contrapositive of the statement, if $\mathcal{Q}'$ does not exist, then no such schedule exists for $\mathcal{Q}$. We show that $\mathcal{Q}'$ can be constructed by decomposing $\mathcal{Q}$ into $\Theta(n)$ subproblems, each of which can be solved in time independent from $n$, thereby providing an algorithm that is tractable with respect to $n$.

Without loss of generality, we assume that $p \cdot n$ is an integer and that $n$ is divisible by $n_0 = \frac{2T}{p}$ for reasons we will explain later. First, we partition the query stream $\mathcal{Q}$ into $d = \frac{n}{n_0}$ disjoint blocks $\mathcal{Q}_1, \ldots, \mathcal{Q}_d$ that are contiguous with respect to arrival time.

Let this partition satisfy the constraint that for any $i < j$, every query in $\mathcal{Q}_i$ has an equal or earlier arrival time than every query in $\mathcal{Q}_j$. For each block $\mathcal{Q}_k$, we remove $2T$ queries with the latest arrival times to form the reduced block $\mathcal{Q}'_k$. Define $\mathcal{Q}' = \bigcup_{k=1}^{d} \mathcal{Q}'_k$.

First, note that if there exists a feasible schedule for $\mathcal{Q}'$ under a uniform TTFT constraint $T$, then there exists a schedule for $\mathcal{Q}$ where the $(1-p)$-th percentile TTFT is at most $T$. This schedule can be constructed by following the schedule for all queries in $\mathcal{Q}'$ and then processing the remaining queries afterwards. By the contrapositive statement, if there does not exist such a schedule for $\mathcal{Q}'$, then there does not exist such a schedule for $\mathcal{Q}$.

We will next show that it is possible to efficiently compute such a schedule for $\mathcal{Q}'$ or certify that none exists due to the decomposable nature of the problem. Since $\mathcal{Q}'_k$ was constructed by removing the last $2T$ queries of $\mathcal{Q}_k$, the latest query arrival time in $\mathcal{Q}'_k$ must be at least $T$ units of time before the arrival time of any query in $\mathcal{Q}'_{k+1}$ if a schedule for $\mathcal{Q}$ satisfying the uniform TTFT constraint exists. This is because no query is an exact prefix of another, and so it must take at least one unit of time to process a query. Therefore, if all queries in $\mathcal{Q}_k \setminus \mathcal{Q}'_k$ satisfy the TTFT constraint, then their arrival times must span at least a $T$ length interval of time.

The partition $\{\mathcal{Q}_k\}_{k \in [d]}$ was constructed to partition the arrival times of $\mathcal{Q}$ into $d$ contiguous intervals. Then, the previous argument implies that the queries in $\mathcal{Q}'_k$ must finished being processed before the earliest arrival time in $\mathcal{Q}'_{k+1}$, and so the processing time of queries in $\mathcal{Q}'$ (the earliest arrival time to the latest completion time) under a processing order of $\mathcal{Q}$ satisfying constraint $T$ can be partitioned into $d$ contiguous time windows, each corresponding to a $\mathcal{Q}'_k$ block. However, this does

not completely decompose the problem, as the processing order of $\mathcal{Q}'_k$ may still affect the the next block $\mathcal{Q}'_{k+1}$ through prefix reuse.

We may handle this dependency by keeping track of the feasible last queries for each block that are possible under schedules that satisfy the TTFT constraint, as these dictate the potential cache states when computing the next block. More concretely, if there is a feasible schedule for $\mathcal{Q}$, then the following procedure must return a feasible schedule for $\mathcal{Q}'$:

1. Let the possible cache initialization of $\mathcal{Q}'_k$ be the feasible end queries of $\mathcal{Q}'_{k-1}$ or the empty string if $k = 1$. Then, in order of $k = 1, ..., d$, compute all possible pairs of a cache initialization and last query processed in $\mathcal{Q}_k$ where a feasible schedule satisfying the constraint $T$ exists.

2. Consider the queries as vertices in a graph along with a vertex representing the empty string initialization and the set of pairs computed in the last step as directed edges in this graph. Compute a path from the empty string vertex to a vertex representing a query in $\mathcal{Q}'_d$.

3. There exists a feasible a schedule for $\mathcal{Q}'$ where the latest processed queries in each $\mathcal{Q}'_k$ is provided by the path computed in the last step. Hence, we may compute schedule for each $\mathcal{Q}'_k$ with the fixed last query with the constraint that the cache initialization and last query processed is dictated by the returned path.

Note that in step one above, if the query in $\mathcal{Q}'_{k-1}$ that is processed latest under the feasible schedule for $\mathcal{Q}$ is recorded as a possible initialization of $\mathcal{Q}'_k$, then the procedure will correctly identify the last query in $\mathcal{Q}'_k$ processed under the feasible schedule for $\mathcal{Q}$ as a possible initialization for $\mathcal{Q}'_{k+1}$. Since the only feasible initialization of the cache at $\mathcal{Q}'_1$ is the empty string, by induction we conclude that step one correctly identifies tuples of feasible cache initialization and end queries for block. From this, there must exist a path returned by step two, since the sequence of tuples that occur correspond to the feasible schedule for $\mathcal{Q}$ must be a directed path from the empty string vertex to the last query processed under the schedule.

Next, we show that the above procedure runs in $\mathcal{O}(n \cdot \exp(1/p \cdot \log 1/p))$ time for each of the above steps:

1. To compute the set of tuples for $\mathcal{Q}'_k$, we must consider at most $n_0$ possible cache initialization and $n_0!$ orderings of queries in $\mathcal{Q}'_k$. We may verify if a fixed combination satisfies the constraint in $\mathcal{O}(n_0)$ time. Therefore, computing set of tuples for a fixed $k$ has the following time complexity:

$$\mathcal{O}(n_0^2 \cdot n_0!) = \mathcal{O}(n_0^{n_0+2}) = \exp(n_0 \cdot \log n_0) = \exp(1/p \cdot \log 1/p).$$

Performing this procedure for each $k$ then takes $\mathcal{O}(n \cdot \exp(1/p \cdot \log 1/p))$ time.

2. For calculating the path, each directed edge is either between a query in $\mathcal{Q}'_k$ to a query in $\mathcal{Q}'_{k+1}$ or from the empty string vertex to a query in $\mathcal{Q}'_1$. Hence, the directed graph is acyclic, and the degree of every vertex is at most $n_0$. Therefore, finding a path from the empty set vertex to a vertex corresponding to a query in $\mathcal{Q}'_d$ can be accomplished by BFS in $\mathcal{O}(n \cdot \mathrm{poly}(n_0))$ time.

3. Finally, given a feasible sequence of last processed queries for each $\mathcal{Q}'_k$, we may compute the schedule for the other queries in $\mathcal{Q}'_k$ while considering a fixed cache initialization and last processed query. This can be done by evaluating all possible schedules in $\mathcal{O}(n_0!) = \exp(1/p \cdot \log 1/p)$ time. Doing this for all $d$ blocks then takes $\mathcal{O}(n \cdot \exp(1/p \cdot \log 1/p))$ time.

Note that computing the partition $\{\mathcal{Q}_k\}_{k \in [d]}$ depends solely on the arrival times and so it can be done in $\mathcal{O}(n)$ time. Finally, to complete the proof, note that we do not need to remove the $2T$ last queries in $\mathcal{Q}_d$ to construct $\mathcal{Q}'_d$, as no other blocks will become dependent on it. Hence, we may adjust the argument by a constant factor, and the assumption that $n$ is exactly divisible by $\frac{2T}{p}$ is not needed. $\quad\square$

# B  Proofs for Section 4

## B.1  Proof of Theorem 2

*Proof.* **LPM:** Label the queries so that $1, 2, \ldots, n$ are in ascending order of arrival times, i.e. $t_1 \leq t_2 \leq \cdots \leq t_n$. Let $\sigma(\cdot)$ be the permutation specifying the *processing order* under LPM. We show that with high probability, *some* query among the *earliest arrivals* (say indices $[q]$) is processed in a *very late position* ($> j$).

Concretely, for integers $q < j$, define the event
$$\{\exists i \in [q] : \sigma(i) > j\} \;=\; \{\text{some earliest-}q \text{ arrival is not processed among the first } j \text{ positions}\}.$$
Since the processing order of LPM with uniformly random tie breaking is independent from the arrival times, $\sigma$ is a uniform random permutation of $[n]$. A standard combinatorial bound then gives:
$$\mathbb{P}\Big( \forall i \in [q], \, \sigma(i) \leq j \Big) \;\leq\; \prod_{r=0}^{q-1} \frac{j-r}{n-r} \;\leq\; \Big(\frac{j}{n}\Big)^q.$$
We set $q = n^{3/4}$ and $j = n - n^{1/2}$. Then,
$$\begin{aligned}
\mathbb{P}(\exists_{i \in [q]} \, \sigma(i) > j) &\geq 1 - \Big(\frac{n - n^{1/2}}{n}\Big)^{n^{3/4}} \\
&= 1 - (1 - 1/n^{1/2})^{n^{3/4}} \\
&= 1 - ((1 - 1/n^{1/2})^{n^{1/2}})^{n^{1/4}}.
\end{aligned}$$
By the known limit $\lim_{x \to \infty}(1 - \frac{1}{x})^x = \frac{1}{e}$, we can conclude that there exists $n_0 \in \mathbb{N}$ such that, for any $n \geq n_0$, $\mathbb{P}(\exists_{i \in [q]} \, \sigma(i) > j) \geq 1 - \delta$.

For a fixed ordering $\mathbf{x}_{i_1}, \ldots, \mathbf{x}_{i_n}$, the time at which the $j$-th query is finished being processed can be written as:
$$\mathtt{TTFT}_{i_j} + t_{i_j} = T + \lceil j/k \rceil \cdot u + d \cdot j, \tag{1}$$
since under LPM, $\lceil j/k \rceil$ unique user prefixes and $j$ unique document suffixes will be processed.

If the event occurs, then there must be a query of index $i \in [q]$ that has not been processed at time $T + \lceil j/k \rceil \cdot u + d \cdot j$. Since $t_i \leq q \cdot s$ for all $i \in [q]$, this implies there exists $i \in [q]$ such that:
$$\begin{aligned}
\mathtt{TTFT}_i &\geq T + \lceil j/k \rceil \cdot u + d \cdot j - q \cdot s \\
&\geq T + j(u/k + d) - qs
\end{aligned}$$
By substituting in the values $q = n^{3/4}$ and $j = n - n^{1/2}$, we conclude the theorem bound for LPM in the theorem statement.

**FCFS:** Label the queries so that $1, 2, \ldots, n$ are in ascending order of arrival times, i.e. $t_1 \leq t_2 \leq \cdots \leq t_n$. FCFS processes these queries in the order $1, 2, \ldots, n$.

Under the $c_{\text{attn}} = 0$, query $i$'s computational time equals:
$$\begin{cases} (u + d), & \text{if the user prefix differs from that of query } i - 1, \\ d, & \text{if query } i \text{ has the same user prefix as query } i - 1. \end{cases}$$
(For $i = 1$, there is no previous query, so the time cost is always $u + d$.)

Define an *indicator* variable
$$I_i \;=\; \begin{cases} 1, & \text{if queries } i \text{ and } i - 1 \text{ share the same user prefix,} \\ 0, & \text{otherwise.} \end{cases}$$

Note that $\mathbb{E}[I_i] = \frac{k-1}{n-1}$ for all $i = 2, \ldots, n$, since the probability that the $i$ and $(i-1)$-th queries share the same (`user`) prefix is $\frac{k-1}{n-1}$. Then, the time needed to process the entire queue is given by:
$$\begin{aligned}
\mathtt{TTFT}_n &= T + |\mathbf{x}_1| + \sum_{i=2}^{n}(|\mathbf{x}_i| - uI_i) - t_n \\
&= T + n(u + d) - sn - u\sum_{i=2}^{n} I_i.
\end{aligned}$$

Markov's inequality states that $\mathbb{P}(X \geq a) \leq \frac{\mathbb{E}[X]}{a}$, where $X$ is a non-negative value and $a > 0$. Applying this to $X = \sum_{i=2}^{n} I_i$ with respect to randomness in the the queue order implies:

$$\mathbb{P}\Big( \sum_{j=2}^{n} I_i \geq \sqrt{n} \Big) \leq \frac{(k-1)\sqrt{n}}{n-1} \leq \frac{k}{\sqrt{n}}.$$

This implies that as $n \to \infty$, $\frac{1}{n} \sum_{i=2}^{n} I_i$ converges to zero in probability. Then the formula can be written as:

$$\texttt{TTFT}_n \geq T + n(u + d - s - \epsilon_n'),$$

where $\epsilon_n'$ converges to zero in probability as $n \to \infty$. By the relation $\max_{i \in [n]}(\texttt{TTFT}_i) \geq \texttt{TTFT}_n$, we conclude the lower bound for the FCFS algorithm in the theorem statement.

$k$-**LPM:** First, note that all queries in the queue must finish being processed by $t = T + n(\frac{u}{k} + d)$. Additionally, the $i$-th query must be processed by time $T + i(u + kd)$, since this is the time needed to complete $i$ groups of user queries, and hence complete $i$ FCFS steps after the start time $T$.

$$\texttt{TTFT}_i \leq \min\{T + i(u + kd) - t_i,\ T + n(\frac{u}{k} + d) - t_i\}$$
$$= \min\{T + i(u + kd) - i \cdot s,\ T + n(\frac{u}{k} + d) - i \cdot s\}$$
$$= \min\{T + i(u + kd - s),\ T + n(\frac{u}{k} + d) - is\}.$$

We want a uniform bound for all $i \in [n]$. Hence, we may bound over the maximum of all indices and then relax the domain to the entire real line.

$$\texttt{TTFT}_i \leq \max_{i \in [n]} \min\{T + i(u + kd - s),\ T + n(\frac{u}{k} + d) - is\}$$
$$\leq \max_{i \in \mathbb{R}} \min\{T + i(u + kd - s),\ T + n(\frac{u}{k} + d) - is\}.$$

Observe that $f_1(i) = T + i(u + kd - s)$ is increasing in $i$, while $f_2(i) = T + n(\frac{u}{k} + d) - is$ is decreasing in $i$. The maximum of $\min\{f_1(i), f_2(i)\}$ occurs where these two lines intersect. We solve

$$i(u + kd - s) = n(\frac{u}{k} + d) - is,$$

which implies,

$$i(u + kd) = n(\frac{u}{k} + d).$$
$$\Rightarrow i^* = \frac{n(\frac{u}{k} + d)}{u + kd} = \frac{n}{k}.$$

Plugging $i^*$ into $f_1$ and $f_2$ yields

$$f_1(i^*) = \frac{n}{k}(u + kd - s), \quad f_2(i^*) = \frac{n}{k}(u + kd - s),$$

so

$$\max_{i \in \mathbb{R}} \min\{f_1(i), f_2(i)\} = \frac{n}{k}(u + kd - s).$$

Hence for all $i \in [n]$,

$$\texttt{TTFT}_i \leq \max_{i \in [n]} \min\Big\{ i(u + kd - s),\ n\big(\frac{u}{k} + d\big) - i\,s \Big\}$$
$$\leq \frac{n}{k}(u + kd - s).$$

$\square$

## B.2    Proof of Corollary 3

*Proof.* From the conditions that $k > 1$ and $s > 0$, it follows that:

$$T + n(\frac{u}{k} + d - \frac{s}{k}) < T + n(\frac{u}{k} + d).$$

Furthermore, from the condition that $u > s$, it follows that:

$$T + n(\frac{u}{k} + d - \frac{s}{k}) < T + n(u + d - s).$$

Hence, by Theorem 2, the corollary statement holds when $c_{\text{attn}} = 0$. We extend to all $c_{\text{attn}} > 0$ by observing that $|\mathbf{x}_i| = u + d$ is constant under Definition 4. Therefore, the time needed to process a query of the regular arrival shuffled queue under Definition 2 with order indexed by $j > 1$ is:

$$(1 + c_{\text{attn}}|\mathbf{x}_j|)(|\mathbf{x}_j| - \text{Overlap}(\mathbf{x}_j, \mathbf{x}_{j-1}))$$
$$= (1 + c)(|\mathbf{x}_j| - \text{Overlap}(\mathbf{x}_j, \mathbf{x}_{j-1})),$$

for some constant $c$ that depends only on $u$ and $d$. Rescaling $s$ by this $1 + c$ term extends the result to all $c_{\text{attn}} > 0$ and concludes the proof.

□

# C  Additional experiment results

## C.1  Real prompt distribution latency metrics

In this section, we provide additional plots for latency metrics corresponding to the experiment of Figure 2.

Note that the relative performance of the scheduling algorithms is accurately predicted by Theorem 2 and surrounding discussion. That is, LPM (equivalent to $k$-LPM with $k = \infty$) attains the best median TTFT as it maximizes prefix reuse. However, $k$-LPM with intermediate values of $k$ attains nearly the same median TTFT and better P99 TTFT by ensuring that no queued query is waiting for too long.

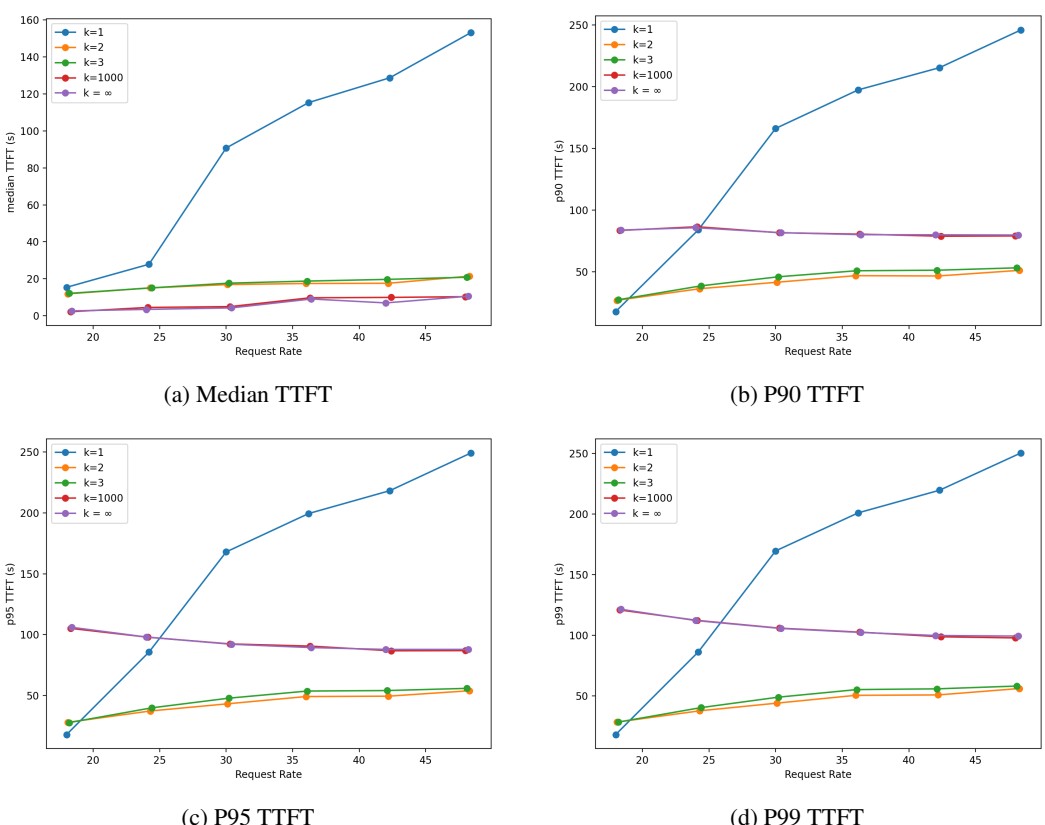

Figure 3: We measure P50, P90, P95, and P99 TTFT versus request rate for five values of the hyperparameter $k$ on 2000 randomly shuffled prompts from the usecase described in Team [2025]. Note that $k = 1$ corresponds to FCFS and $k = \infty$ corresponds to LPM.

## C.2 Varying prefix length ratio

In this section, we provide experiments which vary $u$ and $d$ in the Regular Arrival Shuffled Queue of Definition 4[5]. We set $n = 2000$, $k = 4$ (the number of replications per prefixes), and measure at 80 requests per second. For this experiment, we use a single H100 GPU and the Meta-Llama-3.2-1B-Instruct model Dubey et al. [2024].

Table 3 shows latency results for $k$-LPM with $k$ equal to 1, 4, and $\infty$ respectively, while varying the ratio of the replicated prefix to the constant prompt length (see Definition 4). These results match the relative performance of the settings of $k$ predicted by Theorem 2. That is, LPM consistently results in lower median TTFT due to greater prefix reuse. However, when the ratio of the prefix length to the prompt length ($u$ to $u + d$) is small, FCFS achieves lower TTFT at high percentiles by avoiding any query from waiting for too long. As the the ratio of the prefix length to the prompt length increases, the impact of better prefix reuse becomes greater, and so LPM performs better even regarding higher percentile TTFT. Finally, $k$-LPM with intermediate $k$ never performs simultaneously worse than FCFS and LPM, while it performs better than both in the intermediate regime.

Table 1: TTFT timings in seconds while varying the ratio of the repeated prefix to a constant total string length.

(a) FCFS

| $u$ | $d$ | Median | P90 | P95 |
|------|------|---------|---------|---------|
| 1000 | 5000 | 3466.21 | 5604.33 | 5843.80 |
| 3000 | 3000 | 509.93 | 1040.45 | 1081.62 |
| 5000 | 1000 | 20.15 | 524.91 | 564.44 |

(b) k-LPM

| $u$ | $d$ | Median | P90 | P95 |
|------|------|---------|---------|---------|
| 1000 | 5000 | 1940.35 | 8610.27 | 9128.23 |
| 3000 | 3000 | 177.40 | 821.41 | 1647.49 |
| 5000 | 1000 | 19.61 | 273.55 | 447.37 |

(c) LPM

| $u$ | $d$ | Median | P90 | P95 |
|------|------|---------|---------|---------|
| 1000 | 5000 | 677.07 | 11 297.53 | 14 733.45 |
| 3000 | 3000 | 176.32 | 1245.15 | 2821.72 |
| 5000 | 1000 | 20.70 | 216.88 | 565.92 |

---

[5]Note that we generate the prefix and suffix as random strings of the specified length, which results in non-zero negligible reuse between unique prefix or suffix substrings.

