# OpenReview forum: "LLM Query Scheduling with Prefix Reuse and Latency Constraints"
_NeurIPS.cc/2025/Conference — NeurIPS 2025 poster_

### Official Review · Reviewer_erQW · 2025-06-25

**Clarity:** 2
**Significance:** 2
**Originality:** 3
**Rating:** 4
**Confidence:** 2

**Summary:**

This paper addresses the problem of online query scheduling for LLM inference in systems using prefix reuse, such as RadixAttention. The authors analyze limitations of First Come First Serve (FCFS) and Longest Prefix Match (LPM) schedulers under latency constraints (particularly TTFT) and formally establish the NP-hardness of the scheduling problem. They propose a new scheduling algorithm, k-LPM, that interpolates between FCFS and LPM, and demonstrate both theoretical improvements under a realistic generative model and empirical gains (e.g., P99 TTFT reductions) on Llama-3.1-8B-Instruct using the SGLang serving framework.

**Questions:**

- How sensitive is the performance of `k-LPM` to different prompt distributions, particularly those with low or noisy prefix overlap?
- What is the runtime overhead of computing the k-LPM schedule compared to FCFS or LPM?

**Ethical Concerns:**

["NO or VERY MINOR ethics concerns only"]

**Final Justification:**

The paper proposes a hybrid method that combines FCFS and LPM, achieving a notable reduction in TTFT. The authors provide a rigorous theoretical analysis in a simplified setting and prove that their approach is never worse than either FCFS or LPM (I don't find any problems related to the theorem). I consider this a meaningful and well-justified theoretical contribution.

That said, my recommendation remains a weak accept due to the following limitations:

The theoretical guarantees are established under a simplified problem formulation, which may limit applicability to more complex or realistic scenarios.

In practice, systems often adopt more sophisticated strategies beyond pure FCFS or LPM, which reduces the practical relevance of the baseline comparison.

**Limitations:**

yes

**Quality:**

3

**Strengths And Weaknesses:**

Strengths

- **Problem Formalization.** The paper presents a well-defined and structured model for query scheduling with prefix reuse under latency constraints, and establishes the NP-hardness of the problem via a reduction from 3-PARTITION. This offers a theoretical foundation for understanding the complexity of prefix-aware scheduling in LLM inference.
- **Theoretical Analysis of k-LPM.** The analysis of the proposed `k-LPM` scheduling strategy under a structured generative model (the regular arrival shuffled queue) appears mathematically sound. Theoretical results suggest that `k-LPM` can achieve improved worst-case latency compared to both FCFS and LPM in this setting.
- **Empirical Validation.** The authors support their theoretical claims with experiments on real LLMs and infrastructure. Evaluation on Llama-3.1-8B-Instruct using the SGLang framework demonstrates that `k-LPM` offers tangible improvements in P99 TTFT across a range of request rates, indicating its practical value.

Weaknesses

- **Simplifying Assumptions.** The theoretical model introduces several simplifications—for example, assuming a batch size of one and caching only the most recently processed prompt—which may limit how well the results generalize to production systems with more complex memory and batching behaviors.

---

> ### Author Rebuttal · Authors · 2025-07-31
>
> Thank you for providing this review. Please see our responses below.
>
> > Simplifying Assumptions. The theoretical model introduces several simplifications—for example, assuming a batch size of one and caching only the most recently processed prompt—which may limit how well the results generalize to production systems with more complex memory and batching behaviors.
>
> We make such theoretical assumptions to formalize the problem in a way that allows for clean analysis of the core behavior. While it is possible to assume more complex memory/batching behavior, this would introduce dependency on many hyperparameters (e.g., how long are the prompts, how large is the GPU memory, etc.) which would obscure key ideas. Furthermore, LLM serving frameworks do not have unified behavior that we can model, e.g., treatment of prefix reuse within a batch.
>
> Instead, we structure our paper to theoretically explore the behavior of prefix reuse vs. arrival ordering on tail-end latency under the simplified model. Then, we experimentally validate that these behaviors do indeed hold in a popular production-grade serving framework.
>
> > How sensitive is the performance of k-LPM to different prompt distributions, particularly those with low or noisy prefix overlap?
>
> Theorem 2 of our paper describes rigorously how varying prefix overlap, query length, and arrival time affect expected behavior of $k$-LPM, FCFS, and LPM. For very low prefix reuse, FCFS becomes advantageous and for very high arrival rate, LPM becomes advantageous. The general $k$-LPM method generalizes and interpolates these two extremes. However, $k$-LPM will never be simultaneously worse than FCFS ($k = 1$) or LPM ($k = \infty$), and it will be better in the intermediate regime. In summary, $k$-LPM with intermediate values of $k$ is actually more "robust" than pure FCFS or LPM across variations in these parameters.
>
> Minor noise in the prefix overlap which appears in real world settings does not materially affect the behavior in Theorem 2, since the bulk behavior will be the same. This noise is modeled by the random choice selection use of LPM (see Footnote 4).
>
> > What is the runtime overhead of computing the k-LPM schedule compared to FCFS or LPM?
>
> The additional scheduling overhead of $k$-LPM over LPM is to additionally maintain an ordering with respect to arrival time. While this could be done very efficiently by maintaining pointers between the arriving queries (in order) and the prefix-overlap sorted list, this efficiency is not needed. Sorting a random list of 10000 floats in Python on a CPU takes just 0.006 seconds—essentially nothing compared to typical query‐processing times. Moreover, we can tap into highly optimized C++ sorting libraries to eliminate even that minimal overhead.

---

> > ### Comment · Reviewer_erQW · 2025-08-01
> >
> > Thank you for the rebuttal. After reviewing the paper and the authors’ response, I find that the theoretical contribution is meaningful and the empirical results are promising. I will keep my original score.

---

### Official Review · Reviewer_a5sr · 2025-06-28

**Clarity:** 2
**Significance:** 3
**Originality:** 3
**Rating:** 4
**Confidence:** 3

**Summary:**

This paper presents a formal theoretical framework for scheduling large language model (LLM) queries, particularly in online settings where inference performance needs to be optimized under strict latency constraints like time-to-first-token (TTFT). The research focuses on "prefix reuse", a technique that leverages shared prefixes across queries to reduce computational overhead. The authors reveal limitations of existing scheduling strategies like first-come-first-serve (FCFS) and longest-prefix-match (LPM) and establish that the scheduling problem with prefix reuse under TTFT constraints is NP-Hard. To address this, they propose a novel scheduling algorithm called k-LPM, which generalizes existing methods by balancing prefix reuse and fairness in query processing. The paper provides theoretical guarantees that k-LPM improves TTFT performance under realistic traffic patterns and validates these findings with empirical evaluations showing significant reductions in P99 TTFT compared to baseline methods.

**Questions:**

1. For emerging inference traffic such as long CoT reasoning and deep research, TTFT is not as important as the real-time serving scenarios. Are there any plans to extend the framework to handle such scenarios? How would k-LPM perform in these cases?

**Ethical Concerns:**

["NO or VERY MINOR ethics concerns only"]

**Final Justification:**

I would like to thank the author's rebuttal response and keep my original scores. The experimental part of this paper can be further strengthened, but I still support the acceptance of this paper due to its theoretical contribution.

**Limitations:**

yes

**Quality:**

2

**Strengths And Weaknesses:**

### Strengths

1. Solid Theoretical Foundation: The paper introduces a formal theoretical framework for LLM query scheduling, which provides a robust and well-defined foundation for their analysis.

2. The paper proposes k-LPM, a new scheduling algorithm that generalizes existing approaches like FCFS and LPM, offering a balanced solution for prefix reuse and fairness. Empirical validation shows that k-LPM can reduce the P99 TTFT compared to existing methods.

### Weaknesses

1. My major concern is that the paper presents very little empirical evidence to support its claims. While the theoretical analysis part is strong, the experiments are only limited to one model and one metric (P99 TTFT) in Figure 2. What about other models such as long CoT reasoning models? What about the latency per output token, average TTFT etc? Just to name a few.

2. Online serving systems usually deploy abundant capacity than estimated traffic, while the proposed k-LPM algorithm seems to only benefit when the traffic is high. This would make the proposed algorithm less practical.

---

> ### Author Rebuttal · Authors · 2025-07-31
>
> Thank you for providing this review. Please see our responses below.
>
> > My major concern is that the paper presents very little empirical evidence to support its claims. While the theoretical analysis part is strong, the experiments are only limited to one model and one metric (P99 TTFT) in Figure 2. What about other models such as long CoT reasoning models? What about the latency per output token, average TTFT etc? Just to name a few.
>
> The purpose of our experiments is to show that the theoretical prediction of our formal model holds even under the nuances present in real-world serving frameworks (e.g., dynamic batch size, fixed memory budget, etc.).
>
> The choice of model should not materially affect our described results as long as it fits under the computational model described in Definition 2. Choosing a different model would change the memory needed to store the KV-cache and the amount of time needed for a forward pass which would change the constant scaling with respect to arrival time and prompt length, but not the core behavior described in our theory.
>
> We have conducted additional experiments using the SGLang serving framework by generating synthetic data matching Definition 4 in our paper so that we can systematically control this prompt structure. We will add the full results to our camera ready version, but as an example we observe on a Llama-3.2 1B model that with ($u=4000$, $d=2000$, $k=4$, request\_rate=48), the P99 TTFT of $k$-LPM, FCFS, and LPM is [326, 456, 340] ms respectively, showing that the advantage of $k$-LPM holds across different prompt distributions and models.
>
> Regarding other metrics: TPOT is not generally impacted by prefix reuse since prefix reuse could only have effect if the entire prompt was duplicated, hence we omit this metric. We will add the P50 behavior, and we note that it behaves as theoretically expected. Namely, it is in minimized by $k$-LPM with large $k$ since this maximizes prefix reuse, but this doesn't lead to lower P99 at low request rates as Figure 2 shows (and Theorem 2 predicts).
>
>
> > Online serving systems usually deploy abundant capacity than estimated traffic, while the proposed k-LPM algorithm seems to only benefit when the traffic is high. This would make the proposed algorithm less practical.
>
> While it is true that a server must not be in the high traffic regime on average (otherwise the TTFT would diverge towards infinity), robustness to high traffic is an important property of real serving systems. Tail-end latency is a key quality metric for online systems, and this is roughly equivalent to studying it's behavior in high-traffic periods studied here. In practice, tail-end latency from traffic spikes can be much larger than average latency, and the $k$-LPM algorithm allows for improved robustness, mitigating the impact of these spikes.
>
> > For emerging inference traffic such as long CoT reasoning and deep research, TTFT is not as important as the real-time serving scenarios. Are there any plans to extend the framework to handle such scenarios? How would k-LPM perform in these cases?
>
> Generally, the answer to this question heavily depends on the serving framework and what specifically are the "scheduled" queries. If the "query" being scheduled is the initial prompt that is immediately followed by decoding steps for all reasoning and answer tokens, then it is equivalent to a non-reasoning LLMs from a scheduling perspective. Alternatively, if reasoning is split into separate "queries" to an LLM, then $k$-LPM would automatically account for reuse between the reasoning steps, and each step would behave as a non-reasoning LLM. We are happy to dive into detail for any particular setup you have in mind.

---

> > ### Comment · Reviewer_a5sr · 2025-08-04
> >
> > Thanks for the clarification. I believe this paper can be impactful for LLM serving if it is more broadly tested in production scenarios. I lean for acceptance, but more empirical evaluation would be a plus. I'd like to maintain my score.

---

### Official Review · Reviewer_9K3f · 2025-06-29

**Clarity:** 3
**Significance:** 3
**Originality:** 4
**Rating:** 5
**Confidence:** 2

**Summary:**

The authors study the query scheduling (with prefix reuse) for LLM's problem. They establish that deciding if there is a processing order in query stream such that a time to first token (TTFT) constraint is satisfied is a NP-hard problem. A new query scheduling algorithm k-LPM is introduced that generalizes and outperforms prior methods. Empirical validation of this is carried out on the Llama-3.1-8B-Instruct model.

**Questions:**

* Can the authors discuss the challenges/difficulties of studying the problem with a time per output token constraint? Can techniques in this paper be used to extend results to this case?
* In the experimental section, can the authors add a more thorough description of the data used? In particular, it is not easy to discern the structure of the prompts used in the current format.

**Ethical Concerns:**

["NO or VERY MINOR ethics concerns only"]

**Final Justification:**

The authors addressed all points during rebuttal

**Limitations:**

The authors could consider adding a limitations section

**Quality:**

4

**Strengths And Weaknesses:**

**Strengths**

* A novel theoretical definition of completion time $R(j_k)$ that accounts for query size, size of re-used prefix, and the computational cost of a forward pass through the model architecture.
* A clear and intuitive proof of Theorem 1 that reduces the query scheduling problem to the 3-partition problem.
* The work identifies clear limitations of the LPM/FCFS methods for scheduling and also demonstrates that their method $k-LPM$ can alleviate these issues.

**Weaknesses**

* Experiments are limited to one model and one set of prompts.

---

> ### Author Rebuttal · Authors · 2025-07-31
>
> Thank you for providing this review. Please see our responses below.
>
> > Experiments are limited to one model and one set of prompts.
>
> We agree that evaluating the behavior on additional prompt distributions would be useful. Hence, we have conducted additional experiments using the SGLang serving framework by generating synthetic data matching Definition 4 in our paper so that we can systematically control this prompt structure. We will add the full results to our camera ready version, but as an example we observe on a Llama-3.2 1B model that with ($u=4000$, $d=2000$, $k=4$, request_rate=48), the P99 TTFT of $k$-LPM, FCFS, and LPM is [326, 456, 340] ms respectively, showing that the advantage of $k$-LPM holds across different prompt distributions and models.
>
>
> > Can the authors discuss the challenges/difficulties of studying the problem with a time per output token constraint? Can techniques in this paper be used to extend results to this case?
>
> From a theoretical viewpoint, the difficulty in studying TPOT is that the number of output tokens is random and dependent on input text and conditional probability distribution represented by the LLM. Handling this requires simplifying assumption, for example, as done in [Yang, 2024]
>
> While it may be possible to rigorously study TPOT under our model, it is not high priority for the following reason. Prefix reuse only affects TPOT when an entire prompt overlaps with another and the decoding step leverages this to reduce compute time. Exact prompt matches in a query stream are likely uncommon in practice, and hence existing systems do not optimize for this case.
>
> > In the experimental section, can the authors add a more thorough description of the data used? In particular, it is not easy to discern the structure of the prompts used in the current format.
>
> The prompt structure of our experiments is described at a high-level in lines 264-267, and the referenced source [Team, 2024] provides more in-depth examples of how these prompts look. We will use the extra space in the camera ready version to expand on this explanation to clarify the detailed prompt structure.
>
> > The authors could consider adding a limitations section
>
> We will expand our future work section to both point out limitations in the current analysis and propose potential next steps to address them.

---

> > ### Comment · Reviewer_9K3f · 2025-08-01
> >
> > Thank you for the discussion, I maintain my high score.

---

### Official Review · Reviewer_gHUX · 2025-07-01

**Clarity:** 3
**Significance:** 3
**Originality:** 3
**Rating:** 4
**Confidence:** 2

**Summary:**

The paper formulates the LLM query scheduling problem with a computational model and proves its hardness as an NP-hard problem. The paper further introduces k-LPM —a hybrid scheduler that interpolates the FCFS scheduler (k = 1) and LPM scheduler (k = ∞). Theoretical analysis also shows that k-LPM achieves better TTFT performance than FCFS and LPM on realistic, tree-structured query streams. Experiments on an Llama-3 8B model show that with k = 2 the method significantly reduces P99-TTFT compared to FCFS and LPM under different serving loads.

**Questions:**

Although as shown in the paper, the algorithm is generally robust to the choice of k, I still wonder if there could be some method to choose k principally. For instance, how should k change adaptively as workload changes?

How would your algorithm interact with pre-emptive batching or other priority/fairness scheduling that is widely used?

**Ethical Concerns:**

["NO or VERY MINOR ethics concerns only"]

**Final Justification:**

The reviewer addressed my concerns in the lack of experiments, assumptions, and theories part, and promised to add additional results in the camera ready version. I will maintain my score after the rebuttal.

**Limitations:**

My main concerns are in the experiment sections. Also, see my questions for some further places for improvements.

**Quality:**

3

**Strengths And Weaknesses:**

**Strength**
- This paper targets on an important problem that is highly relevant to the wide application of multi-agent systems and complicated tree-based reasonings.
- This paper gives a theoretical model to describe the scheduling problem under TTFT constraints. The paper further reduces this problem to 3-partition problem to show that this problem is NP-hard. This shed light on future research to find better approximations instead of exact solutions for this problem.
- The proposed method(L-kpm) is intuitive and the author gives theoretical guarntees on why this method is better than FCFS and LPM
- Experiments on Llama-3.1 8b shows that this method can significantly reduce the TTFT over different loads

**Weakness**
- Lack of evaluation on different traces:  The experiment is conducted on a relatively small model and limited workloads with only 2k prompts. More different workloads with different level of reusable prompts are helpful to comprehensively evaluate the performance of k-LPM.
- Lack of evaluation with different metrics: Only the P99 TTFT is reported. What about the P50 TTFT, P95 TTFT?
- The generative model is based on simple assumption “Our data generative model considers the simplest case of prompts constructed from a height two prefix tree, where the edges at each depth are constant”. This may hurt the robustness of k-LPM of the further proofs.
- Although it has been shown in the paper that the overhead is minimal theoretically, it would be better to provide some breakdown experiments on the overhead

---

> ### Author Rebuttal · Authors · 2025-07-31
>
> Thank you for providing this review. Please see our responses below.
>
> > Lack of evaluation on different traces: The experiment is conducted on a relatively small model and limited workloads with only 2k prompts. More different workloads with different level of reusable prompts are helpful to comprehensively evaluate the performance of k-LPM.
>
> We agree that evaluating the behavior at different levels of prefix overlap would be useful. Hence, we have conducted additional experiments using the SGLang serving framework by generating synthetic data matching Definition 4 in our paper so that we can systematically control this prompt structure. We will add the full results to our camera ready version, but as an example we observe on a Llama-3.2 1B model that with ($u=4000$, $d=2000$, $k=4$, request\_rate=48), the P99 TTFT of $k$-LPM, FCFS, and LPM is [326, 456, 340] ms respectively, showing that the advantage of $k$-LPM holds across different prompt distributions and models.
>
>
> > Lack of evaluation with different metrics: Only the P99 TTFT is reported. What about the P50 TTFT, P95 TTFT?
>
> We provide P99 TTFT since our focus is on tail-end latency, which is a critical consideration of online systems. However, we will add graphs for P50 and P95 in the appendix to complement Figure 2. Interestingly, we again see the behavior predicted by our theory. That is, P50 is better for large values of $k$ across all request rates (since prefix reuse is maximized), but this doesn't translate to better P99 TTFT at high request rates as Figure 2 shows (as predicted by Theorem 2).
>
> > The generative model is based on simple assumption “Our data generative model considers the simplest case of prompts constructed from a height two prefix tree, where the edges at each depth are constant”. This may hurt the robustness of k-LPM of the further proofs.
>
> We make these assumptions since it does not materially affect the conclusion of our results while greatly simplifying the notation and exposition, allowing the core ideas to be presented more clearly. We see from our experiments on real world data that these assumptions are not necessary for the predicted behavior to hold.
>
> One can prove an analogous result for any query distribution where queries have variable arrival times and variable prefix overlaps, since the key idea that over-optimizing for prefix reuse results in long tail latency and ignoring prefix reuse result in low throughput still holds. The precise theorem and proof would depend on the parameterization of the query distribution.
>
> > Although it has been shown in the paper that the overhead is minimal theoretically, it would be better to provide some breakdown experiments on the overhead
>
> The additional scheduling overhead of $k$-LPM over LPM is to additionally maintain an ordering with respect to arrival time. While this could be done very efficiently by maintaining pointers between the arriving queries (in order) and the prefix-overlap sorted list, this efficiency is not needed. One can sort a randomly ordered list of 10k floats in Python in 0.006 seconds, which is negligible compared to query processing speed, and so the algorithmic overhead is negligible.
>
> > Although as shown in the paper, the algorithm is generally robust to the choice of k, I still wonder if there could be some method to choose k principally. For instance, how should k change adaptively as workload changes?e
>
> We agree this is an interesting direction to explore, and briefly touch on some potential methods at line 313. Two initial ideas are to use backtesting which works well on stationary data or multi-armed bandit algorithms which could attain optimal asymptotic regret. However, there are many possible approaches (e.g., controller methods) and it is worth exploring further.
>
> > How would your algorithm interact with pre-emptive batching or other priority/fairness scheduling that is widely used?
>
> While it is difficult to answer precisely without knowing the exact rules of the scheduling method, we generally expect that they should be rather orthogonal to $k$-LPM. Pre-emption has the downside that it wastes computation, and hence achieves lower average throughput compared to $k$-LPM, but may enforce a harder constraint on the fairness.
>
> Such ideas may be complementary, since one can use the $k$-LPM algorithm by default to achieve a smooth tradeoff between prefix reuse and prioritizing older queries. Pre-emption can then be used as a layer on top of this to protect against resource starvation of long-waiting queries, though it would be preferable to handle this by setting $k$ optimally instead.

---

> > ### Comment · Reviewer_gHUX · 2025-08-02
> >
> > Thank you for your rebuttal. After reviewing the author's discussion, I think my concerns are broadly addressed regarding the assumptions, theories, and the experiments. I hope the additional experimental details and discussions could be added in camera-ready version.

---

### Decision · Program_Chairs · 2025-09-17

**Decision:**

Accept (poster)

**Comment:**

The paper studies LLM query scheduling, which is an important problem. All the reviewers agree on the paper's solid theoretical contribution. While some reviewers raised concerns about experimental parts, the authors effectively addressed these concerns during the rebuttal.